# Crosstalk among WEE1 Kinase, AKT, and GSK3 in Nav1.2 Channelosome Regulation

**DOI:** 10.3390/ijms25158069

**Published:** 2024-07-24

**Authors:** Aditya K. Singh, Jully Singh, Nana A. Goode, Fernanda Laezza

**Affiliations:** Department of Pharmacology & Toxicology, The University of Texas Medical Branch, Galveston, TX 77555, USA; jusingh@utmb.edu (J.S.); nagoode@utmb.edu (N.A.G.); felaezza@utmb.edu (F.L.)

**Keywords:** voltage-gated sodium channels, fibroblast growth factors, split luciferase assay, kinase inhibitors

## Abstract

The signaling complex around voltage-gated sodium (Nav) channels includes accessory proteins and kinases crucial for regulating neuronal firing. Previous studies showed that one such kinase, WEE1—critical to the cell cycle—selectively modulates Nav1.2 channel activity through the accessory protein fibroblast growth factor 14 (FGF14). Here, we tested whether WEE1 exhibits crosstalk with the AKT/GSK3 kinase pathway for coordinated regulation of FGF14/Nav1.2 channel complex assembly and function. Using the in-cell split luciferase complementation assay (LCA), we found that the WEE1 inhibitor II and GSK3 inhibitor XIII reduce the FGF14/Nav1.2 complex formation, while the AKT inhibitor triciribine increases it. However, combining WEE1 inhibitor II with either one of the other two inhibitors abolished its effect on the FGF14/Nav1.2 complex formation. Whole-cell voltage-clamp recordings of sodium currents (I_Na_) in HEK293 cells co-expressing Nav1.2 channels and FGF14-GFP showed that WEE1 inhibitor II significantly suppresses peak I_Na_ density, both alone and in the presence of triciribine or GSK3 inhibitor XIII, despite the latter inhibitor’s opposite effects on I_Na_. Additionally, WEE1 inhibitor II slowed the tau of fast inactivation and caused depolarizing shifts in the voltage dependence of activation and inactivation. These phenotypes either prevailed or were additive when combined with triciribine but were outcompeted when both WEE1 inhibitor II and GSK3 inhibitor XIII were present. Concerted regulation by WEE1 inhibitor II, triciribine, and GSK3 inhibitor XIII was also observed in long-term inactivation and use dependency of Nav1.2 currents. Overall, these findings suggest a complex role for WEE1 kinase—in concert with the AKT/GSK3 pathway—in regulating the Nav1.2 channelosome.

## 1. Introduction

Voltage-gated sodium channels (Nav1.1–1.9) are composed of the pore-forming α-subunit, which is necessary for ion conduction [1]. Their complete physiological function, however, depends on protein–protein interactions (PPIs) with auxiliary proteins [2,3], including members of the intracellular fibroblast growth factor (iFGF) family, such as FGF14 [4]. Nav1.2 channels are predominant in the central nervous system (CNS) during early childhood, whereas Nav1.6 channels become more prevalent in adulthood [5].

These channels are essential for generating and propagating action potentials in neurons during their respective developmental stages [6,7,8,9]. Notably, FGF14, which is highly expressed in the brain, directly regulates both Nav1.2 and Nav1.6 channels by interacting with their C-terminal domains (CTD) [4], thereby exerting isoform-specific modulatory effects on sodium currents. Studies have demonstrated that kinases play a critical role in regulating the FGF14/Nav1.6 complex through phosphorylation of either the channel itself or FGF14 [4,10,11]. This evidence suggests that FGF14 and various kinases collectively constitute a complex signalosome centered around the intracellular domains of the Nav1.6 channel, which is critical for maintaining its function in neurons. Serine/threonine kinases such as GSK3β [4,10] and CK2 [12], in addition to the tyrosine kinase JAK2 [13], have been identified as pivotal in mediating the Nav1.6 signalosome. However, less is known about the interplay between these kinases and FGF14 in regulating Nav1.2 function—a gap in knowledge that could be particularly relevant in the context of Nav1.2 channelopathies, which have been implicated in the incidence of neurodevelopmental disorders.

In previous studies, we demonstrated that the FGF14/Nav1.2 complex is selectively regulated by WEE1 [14], a dual kinase with serine/threonine and tyrosine catalytic functions [15]. WEE1 is well known for its role in regulating cell cycle checkpoint complexes composed of cyclin-dependent kinases (CDK) and cyclins [16], which controls the entry of cells into mitosis for DNA repair [17,18,19,20,21], a function that has implications in the cancer field [22,23,24,25]. Additionally, WEE1 has been reported to have regulatory epigenetic effects [26,27]. Although not much is known about WEE1 regulatory activity in postmitotic differentiated cells such as neurons, [28,29,30] studies from the cancer field indicate that WEE1 activity is closely linked to AKT [20,31,32,33] and GSK3 [34]. WEE1 can be controlled by GSK3, which regulates its degradation [17,35], and can work synergistically with AKT [20], a well-known upstream suppressor of GSK3 [36,37]. These intricate positive and negative feedback loops between WEE1, GSK3, and AKT may also take place in neurons, indirectly impacting FGF14/Nav1.2 complex formation and Nav1.2 channel activity.

Here, we explored the potential connectivity of WEE1, AKT, and GSK3 on FGF14/Nav1.2 channel complex formation using a split-luciferase complementation assay (LCA), combined with patch-clamp electrophysiology for functional assessment of Nav1.2 currents elicited in the presence of FGF14. Through this approach, we first uncovered new regulatory functions of AKT and GSK3 on the FGF14/Nav1.2 complex and Nav1.2 currents. We then found that the modulatory effects of WEE1 inhibitors were influenced by the presence of AKT and GSK3 inhibitors, suggesting cooperative or competitive effects on FGF14/Nav1.2 complex formation and Nav1.2 currents. Overall, these findings indicate that the FGF14/Nav1.2 signalosome involves a connecting mode of WEE1-dependent AKT/GSK3 signaling pathways. This study could provide insights into the signaling mechanisms underlying neurodevelopmental disorders associated with Nav1.2 channelopathies [38,39,40,41,42,43,44,45], aiding in the development of targeted therapies for these conditions based on modulating Nav1.2 PPIs.

## 2. Results

### 2.1. Pharmacological Interrogation of WEE1, AKT, and GSK3 Inhibitors on the FGF14/Nav1.2 Complex Assembly

To study the pharmacological effects of inhibitors of WEE1, AKT, and GSK3 on the FGF14/Nav1.2 complex assembly, we used the LCA previously optimized for reconstituting the FGF14/Nav1.2 complex in cells [14]. To this end, HEK293 cells were transiently transfected with CD4-Nav1.2-CTD-NLuc and CLuc-FGF14 cDNA constructs, enabling the interaction of FGF14 with the CTD of the Nav1.2 channel. This interaction reconstitutes the NLuc and CLuc fragments of the luciferase enzyme and, in the presence of the substrate luciferin, produces a robust luminescence signal (Figure 1). The LCA signal resulting from FGF14/Nav1.2 complex formation was then evaluated in the presence of WEE1 inhibitor II, the AKT inhibitor triciribine, and the GSK3 inhibitor XIII, both alone and in pairwise combination with the WEE1 inhibitor and compared with the vehicle control group (DMSO 0.5%; Figure 1). While WEE1 inhibitor II and GSK3 inhibitor XIII caused a dose-dependent decrease in the FGF14/Nav1.2 complex assembly (WEE1 inhibitor II IC_50_ = 17.6 µM, Figure 1A; GSK3 inhibitor XIII IC_50_ = 21.1 µM, Figure 1C), triciribine caused a dose-dependent increase in the FGF14/Nav1.2 complex assembly (EC_50_ = 33.6 µM; Figure 1B). To investigate crosstalk among WEE1, AKT, and GSK3 in regulating FGF14/Nav1.2 complex formation, cells were treated first with WEE1 inhibitor II, followed 15 min later by triciribine or GSK3 inhibitor XIII application. Notably, pretreatment of cells with a concentration of WEE1 inhibitor II (15 µM) close to its IC_50_ value allowed it to outcompete the effect of triciribine by shifting its EC_50_ to the right (EC_50_ = 57.8 µM, Figure 1D). On the other hand, pretreatment with WEE1 inhibitor II completely nullified the inhibitory effect of GSK3 inhibitor XIII on the FGF14/Nav1.2 complex assembly, resulting in a luminescence signal that was comparable to the vehicle-treated control group (Figure 1E). Notably, WEE1 inhibitor and GSK3 inhibitor XIII, alone or in combination, were ineffective in regulating the complex assembly in the presence of the FGF14^Y158A^ mutant, suggesting that the effect of these two kinases may converge on the FGF14^Y158A^ residue. The FGF14^Y158A^ mutation did not prevent the effect of triciribine, alone or in combination with WEE1 inhibitor II, on the FGF14^Y158A^/Nav1.2 complex assembly, indicating the involvement of other residues in FGF14 or Nav1.2. (Appendix A). Overall, these data reveal the following: i. both AKT and GSK3 control FGF14/Nav1.2 complex formation, with effects similar in direction and magnitude to those reported for the FGF14/Nav1.6 complex [46]; ii. WEE1 regulation of the FGF14/Nav1.2 complex assembly occurs not only directly as previously reported [14] but also via crosstalk with AKT and GSK3; iii. WEE1 and GSK3, but not AKT, regulate the FGF14/Nav1.2 complex assembly via FGF14^Y158A^.

### 2.2. Functional Regulation of Nav1.2—Mediated Currents through WEE1, AKT, and GSK3 Kinase Crosstalk

In prior research, we demonstrated that WEE1 inhibitor II specifically modulates Nav1.2 currents in an FGF14-dependent manner and that this regulatory effect was contingent on the Y158 amino acid site on FGF14 [14], a crucial site at the PPI interface between iFGFs and Nav channel CTDs. These findings suggest that WEE1, with its dual S/T and Y phosphorylation activity, may directly regulate Nav1.2 currents elicited in the presence of FGF14 through phosphorylation of Y158. In this study, we demonstrate that WEE1 modulation of FGF14/Nav1.2 complex formation involves crosstalk with AKT and GSK3, leading to the hypothesis that this interplay could result in concerted regulation of Nav1.2 currents. To test this, we used whole-cell patch-clamp electrophysiology on HEK293 cells stably expressing Nav1.2 that were transiently transfected with FGF14-GFP (HEK-Nav1.2/FGF14) and treated with either 0.01% DMSO (vehicle, control), WEE1 inhibitor II (15 µM), triciribine (25 µM), or GSK3 inhibitor XIII (30 µM), alone or in paired combinations (Figure 2). In agreement with prior research [14], WEE1 inhibitor II significantly suppressed Nav1.2 transient sodium currents (I_Na_) compared with control FGF14-GFP (WEE1 inhibitor II: −31.34 ± 7.0 pA/pF, *n* = 9 vs. FGF14-GFP (vehicle): −83.85 ± 6.0 pA/pF, *n* = 10, *p* = 0.0008, one-way ANOVA followed by Tukey’s multiple comparisons test). However, unlike treatment with triciribine, which has a mild but insignificant effect on I_Na_ (−78.8 ± 18.2, *n* = 7, *p* = >0.9999; Figure 2A–C), treatment of cells with GSK3 inhibitor XIII significantly increased I_Na_ (−127.6 ± 7.2 pA/pF, *n* = 6, *p* = 0.0292; Figure 2A–C). Interestingly, pretreatment with WEE1 inhibitor II abolished the effect of GSK3 inhibitor XIII on I_Na_ (WEE1 inhibitor II + GSK3 inhibitor XIII: −40.7 ± 4.2 pA/pF FGF14-GFP: −83.85 ± 6.0 pA/pF, *p* = 0.0330, *p* = 0.0001, *n* = 6) and counterbalanced the mild effect of triciribine leading to I_Na_ values comparable to WEE1 inhibitor II alone (WEE1 inhibitor II + triciribine: −26.3 ± 5.5 pA/pF; FGF14-GFP: −83.85 ± 6.0 pA/pF, *p* = 0.0003, *n* = 8; Figure 2A–C). Further analysis revealed that in the presence of WEE1 inhibitor II, the tau of fast inactivation was significantly slower (2.1 ± 0.32 ms) compared with the control (1.21 ± 0.07 ms, *n* = 10, *p* = 0.0103). Co-treatment of WEE1 inhibitor II with triciribine produced a decrease in tau (−2.12 ± 0.27 ms) comparable to the single treatment with WEE1 inhibitor II. Interestingly, while GSK3 inhibitor XIII alone had no effect on the tau of fast inactivation (−1.33 ± 0.07 ms) compared with the control (−1.21 ± 0.07 ms), the combined treatment of WEE1 inhibitor II and GSK3 inhibitor XIII suppressed the effect of WEE1 inhibitor II on fast inactivation kinetics. These results suggest that AKT, GSK3, and WEE1 play distinct roles in regulating Nav1.2 currents and channel kinetics and that while WEE1 regulation of I_Na_ density prevails over AKT and GSK3, its control over fast inactivation is influenced by GSK3.

### 2.3. Nav1.2 Voltage Sensitivity Is Modulated by WEE1 Kinase, AKT, and GSK3 

Given the distinct effects of the three inhibitors on regulating I_Na_ density, likely influenced by Nav1.2 channel trafficking to the plasma membrane, versus kinetic parameters like the tau of fast inactivation, which are dictated purely by channel biophysics, we aimed to investigate further potential differences in the regulatory mechanisms of these three kinases, both alone and in combination, on voltage dependence of activation and steady-state inactivation. As shown in Figure 3, all three inhibitors affected V_1/2_ of activation: WEE1 inhibition caused a depolarizing shift in V_1/2_ of activation compared with the control (WEE1: −17.3 ± 1.0 mV, DMSO: −21.0 ± 0.9 mV, *n* = 10, *p* = 0.0184); triciribine or GSK3 inhibitor XIII caused a hyperpolarizing shift (triciribine: −26.95 ± 1.4 mV, GSK3 inh. XIII −27.75 ± 0.9, *p* = 0.0015 and *p* = 0.0001); and importantly, WEE1 inhibition successfully countered the effect of triciribine but failed to oppose the effect of GSK3 inhibitor XIII (Figure 3A,B; Table 1; Appendix A). Likewise, all three kinase inhibitors shifted the V_1/2_ of steady-state inactivation to a more depolarized level compared with the control. Interestingly, despite both WEE1 and AKT inhibitors having the same directional effect on V_1/2_ steady-state inactivation (depolarizing shift) when tested separately, in combination, they restored V_1/2_ to the control. Furthermore, when co-applied with GSK3 inhibitor XIII, the WEE1 inhibitor induced a depolarizing effect on V_1/2_ inactivation (−55.14 ± 1.1 mV), comparable to the effect observed with the combination (−50.8 ± 1.4 mV) or single treatment (−48.54 ± 0.4 mV). Overall, these results indicate distinct mechanisms by which the three kinases regulate Nav1.2 currents, denoting competitive, convergent, or additive effects depending on the channel’s conformational changes and cycle state.

### 2.4. WEE1 Kinase, AKT, and GSK3 Modulate Long-Term Inactivation and Use Dependency of the Nav1.2 Channel 

Intracellular FGFs are crucial for regulating long-term inactivation (LTI) and use-dependent mechanisms of Nav channels, which govern channel availability during repetitive stimulation [47,48,49]. To investigate the impact of WEE1, AKT, and GSK3 on LTI and cumulative inactivation, HEK-Nav1.2 FGF14 cells were treated with corresponding inhibitors either individually or in paired combinations and then subjected to repetitive pulses of variable duration and frequency to induce LTI or cumulative inactivation through use dependency. When applied alone, WEE1 inhibitor II significantly potentiated Nav1.2 currents opposing any form of LTI or use dependency (Figure 4A–C). Triciribine had different effects on LTI and use dependency, preventing any form of LTI but promoting cumulative inactivation through use dependency (Figure 4D,E). In the combined treatment, WEE1 inhibitor II prevailed over triciribine in both LTI (Figure 4A–C) and use dependency (Figure 4D,E). On the other hand, GSK3 inhibitor XIII had no effects on either LTI (GSK3 inh. XIII 101.1 ± 3.6, DMSO −99.24 ± 1.9) or use dependency compared with the control (GSK3 inh. XIII 1.0 ± 0.04, DMSO −1.07 ± 0.05). Intriguingly, similar to the effects of combined treatment on voltage dependence of activation and steady-state inactivation, the combination of WEE1 inhibitor II and GSK3 inhibitor XIII led to levels of LTI and use dependency that were comparable to the control (Figure 4A–E). All results are detailed in Table 2. Overall, these results suggest that WEE1 and AKT work synergistically in regulating LTI but competitively in regulating use dependency. Conversely, WEE1 and GSK3 compete both in regulating LTI and use dependency. 

## 3. Discussion

In this study, we explored the role of WEE1 kinase in regulating the FGF14/Nav1.2 channel complex, alongside AKT and GSK3, kinases previously linked to WEE1 [20,34]. 

Our results, derived from LCA measurements and various electrophysiological protocols, reveal that WEE1, AKT, and GSK3 interplay in regulating the FGF14/Nav1.2 complex assembly and Nav1.2 currents, indicating pathway competition or synergy for each phenotype measured. Furthermore, this study expands on the importance of iFGFs in priming the Nav channel complex for kinase regulation, contributing to the integrity of the Nav channel signalosome in neurons and highlighting the complexity of intracellular signaling governing neuronal excitability. Previous studies have shown that WEE1 regulation of Nav1.2 is isoform specific and requires FGF14^Y158^ [14], a critical residue at the PPI interface and a site of phosphorylation [13]. Additionally, GSK3β has been shown to directly phosphorylate T1966 on Nav1.2 [50] and S226 on FGF14 [11]. Thus, regulation of the FGF14/Nav1.2 complex assembly and Nav1.2 currents by WEE1 and GSK3 likely involves direct phosphorylation of either FGF14 and/or Nav1.2. We have no evidence that AKT directly phosphorylates FGF14 or Nav1.2. However, AKT inhibits GSK3 (both isoform α and β) via inhibitory phosphorylation at S9/S21 [37], and GSK3 regulates WEE1 through ubiquitination [35]. Additionally, there is evidence for WEE1 inhibitor synergy with AKT inhibitors [20], suggesting a positive feedback loop. Therefore, WEE1 may exert regulatory effects on the Nav1.2 channel through direct regulation or via synergy or competition with the AKT/GSK3 signaling pathway. A schematic of these potential pathways is summarized in Figure 5.

In the LCA, WEE1 inhibitor II, triciribine, and GSK3 inhibitor XIII all influenced FGF14/Nav1.2 complex formation. WEE1 inhibitor II and GSK3 inhibitor XIII suppressed complex formation, while triciribine increased it. When WEE1 inhibitor II was combined with triciribine, triciribine’s effect outcompeted WEE1 inhibition. This can be interpreted as GSK3 disinhibition dominating the phenotype and leading to increased FGF14/Nav1.2 assembly. Conversely, when WEE1 inhibitor II was combined with GSK3 inhibitor XIII, the two treatments canceled each other’s effects. This may result from a net equilibrium between the WEE1 inhibitor suppressing WEE1 activity and the GSK3 inhibitor XIII suppressing GSK3′s inhibitory control over WEE1 through degradation.

In the electrophysiological experiments, all inhibitors influenced Nav1.2 currents, both individually and in combination. WEE1 inhibitor II significantly suppressed I_Na_ density, and its effect prevailed over that of triciribine and GSK3 inhibitor XIII. However, the ability of WEE1 inhibitor II to slow fast inactivation was dominant only over triciribine and was nullified by GSK3 inhibitor XIII. The three inhibitors exhibited distinct effects on the voltage sensitivity of Nav1.2 activation when tested individually. WEE1 inhibitor II caused a depolarizing shift in the V_1/2_ of activation, while triciribine and GSK3 inhibitor XIII induced a hyperpolarizing shift in the V_1/2_ of activation. In combination with triciribine, WEE1 inhibitor II’s depolarizing effect dominated. However, when combined with GSK3 inhibitor XIII, WEE1 inhibitor II was unable to counteract the hyperpolarizing effect of the GSK3 inhibitor. 

The impact of the three inhibitors on the V_1/2_ of steady-state inactivation was consistent when tested individually. WEE1 inhibitor II, triciribine, and GSK3 inhibitor XIII each caused a shift toward a more depolarized level compared with the control. Surprisingly, there was an unexpected cooperation between WEE1 and AKT, as their combined inhibition restored the V_1/2_ to the control level. However, when WEE1 inhibitor II and GSK3 inhibitor XIII were used alone or together, they induced an equal depolarizing shift in the V_1/2_, suggesting potential convergence of these kinases on the same regulatory mechanism. Overall, the net physiological effect of these kinases on Nav1.2 current voltage sensitivity is expected to shift the threshold for eliciting action potentials in neurons, thereby regulating firing and ultimately synaptic transmission.

The effects of the three inhibitors on LTI and use dependency were more complex. WEE1 inhibition led to potentiation of Nav1.2 currents in both protocols, suggesting a role of WEE1 in promoting channel entry into slow inactivation and fast inactivation. This is supported and consistent with WEE1 inhibition slowing the tau of fast inactivation. Conversely, GSK3 inhibition alone did not significantly alter LTI or use dependency. However, in combination, WEE1 inhibitor II and triciribine synergistically regulated LTI while competing in the regulation of use dependency. Conversely, WEE1 inhibitor II and GSK3 inhibitor XIII exhibited mild regulation of both LTI and use dependency, particularly when applied together.

With the exception of the regulation of I_Na_ density, in which WEE1 inhibition appears to prevail over AKT and GSK3, WEE1 kinase and GSK3 appear to directly compete. This competition may be explained by GSK3 inhibitor XIII restoring a pool of active WEE1 by limiting its degradation mediated by GSK3 [35]. On the other hand, WEE1 and AKT either compete, possibly due to GSK3 inhibition conferred by triciribine [37], or synergize through mechanisms similar to those reported in cancer cells [20]. 

Overall, these findings underscore the diverse mechanisms by which the three kinases regulate both the FGF14/Nav1.2 complex assembly and Nav1.2 currents. These mechanisms manifest as either competitive or synergistic interactions, influenced by factors such as the complexity of the system tested (e.g., minimal functional domain in LCA versus full channel in electrophysiology) or the cycle stage of the channel. Future research will elucidate the molecular basis of these regulatory mechanisms driven by WEE1.

Physiologically, Nav1.2 is widely expressed in neurons during embryonic and early-stage development, facilitating action potential backpropagation, synaptic integration, and plasticity [38,51]. As neuronal maturation progresses, the neonatal Nav1.2 isoform [52] is gradually replaced by Nav1.6, which becomes the dominant isoform in adulthood [5]. Because WEE1 exerts specific regulation on Nav1.2 but not Nav1.6 channels, unbalanced levels of WEE1 could perturb the developmental switch between Nav1.2 and Nav1.6 isoforms, delaying or accelerating neuronal maturation with consequences for synaptic integration and plasticity. Both WEE1 and FGF14 have been associated with schizophrenia and other neurodevelopmental disorders [53,54]. Thus, WEE1 may be part of a signaling pathway, including FGF14 and Nav1.2, that, if perturbed, could contribute to endophenotypes related to neurodevelopmental disorders such as schizophrenia and *SCN2A* channelopathies associated with autism spectrum disorder and epilepsy [38,39,40,41,42,43,44,45,55,56,57,58,59,60,61].

## 4. Materials and Methods

### 4.1. DNA Constructs

The CLuc-FGF14, CD4-Nav1.2 CTD-NLuc, and pQBI-FGF14-GFP cDNA constructs were engineered and characterized as previously described [14].

### 4.2. HEK293 Cell Culture

HEK293 cells were cultured and maintained in DMEM and F-12 (Invitrogen, Carlsbad, CA, USA), supplemented with 0.05% glucose, 0.5 mM pyruvate, 10% fetal bovine serum, 100 units/mL penicillin, and 100 µg/mL streptomycin (Invitrogen), and incubated at 37 °C with 5% CO_2_. For transfection, cells were seeded in 24-well CELLSTAR^®®^ tissue culture plates (Greiner Bio-One, Monroe, NC, USA) at 4.5 × 105 cells per well and incubated overnight to reach monolayers with 90–100% confluency. The cells were then transiently transfected with pQBI-FGF14-GFP using Lipofectamine 2000 (Invitrogen), according to the manufacturer’s instructions, using 1 µg of plasmid per transfection per well. HEK293 cells stably expressing the human Nav1.2 channel were maintained similarly, except for the addition of 500 µg/mL G418 (Invitrogen) to maintain stable Nav1.2 expression. Cells were transfected at 80–90% confluence with FGF14-GFP using Lipofectamine 2000 (Invitrogen) according to the manufacturer’s instructions. HEK-Nav1.2/FGF14 cells were washed and re-plated at very low density prior to electrophysiological recordings.

### 4.3. Split-Luciferase Complementation Assay

The split-luciferase complementation assay (LCA) was conducted following established protocols. HEK293 cells were transiently transfected with either the CLuc-FGF14 and CD4-Nav1.2 CTD-NLuc pair of DNA constructs using Lipofectamine 3000 (Invitrogen), following the manufacturer’s instructions. Transiently transfected cells were replated into CELL-STAR μClear^®^ 96-well tissue culture plates (Greiner Bio-One, Monroe, NC, USA) 48 h post-transfection. After 24 h, the medium was replaced with serum-free, phenol red-free, 1:1 DMEM/F12 (Invitrogen) containing WEE1 inhibitor II, AKT inhibitor (triciribine), or GSK3 inhibitor XIII (all purchased from Calbiochem, San Diego, CA, USA) were dissolved in DMSO (1–150 or 0.5–100 µM), or DMSO alone. The final concentration of DMSO was maintained at 0.5% for all wells. Subsequently, after 2 h of incubation at 37 °C, the reporter reaction was initiated by the addition of 100 μL substrate solution containing 1.5 mg/mL D-luciferin (Gold Biotechnologies, St. Louis, MO, USA) dissolved in PBS. Luminescence reaction readings were then performed using a SynergyTM H1 Multi-Mode Microplate Reader (BioTek, Winooski, VT, USA), and acquired data were analyzed as previously described.

### 4.4. Whole-Cell Patch Clamp Electrophysiology

HEK-Nav1.2 cells were transfected with FGF14-GFP and plated at low density on glass coverslips for 3–4 h. Electrophysiological recordings were conducted at room temperature using a MultiClamp 200B amplifier (Molecular Devices, San Jose, CA, USA) after a 60 min incubation with 0.01% DMSO or WEE1 inhibitor II (15 µM), triciribine (25 µM), or GSK3 inhibitor XIII (30 µM) in extracellular solution. The composition of the recording solutions comprised the following salts: extracellular (mM): 140 NaCl, 3 KCl, 1 MgCl2, 1 CaCl2, 10 HEPES, 10 glucose, pH 7.3; intracellular (mM): 130 CH3O3SCs, 1 EGTA, 10 NaCl, 10 HEPES, pH 7.3. Membrane capacitance and series resistance were estimated by the dial settings on the amplifier and electronically compensated for by 70–80%. Data were acquired at 20 kHz and filtered at 5 kHz before digitization and storage. All experimental parameters were controlled by Clampex 9.2 software (Molecular Devices) and interfaced with the electrophysiological equipment using a Digidata 1300 analog–digital interface (Molecular Devices). Voltage-dependent inward currents for HEK-Nav1.2/FGF14 cells were evoked by depolarization test potentials between −100 mV (Nav1.2) and +60 mV from a holding potential of −70 mV. Steady-state (fast) inactivation of Nav channels was measured with a paired-pulse protocol. From the holding potential, cells were stepped to varying test potentials between −120 mV and +20 mV (prepulse) prior to a test pulse to −20 mV.

Current densities were obtained by dividing Na^+^ current (I_Na_) amplitude by membrane capacitance. Current–voltage relationships were generated by plotting current density as a function of the holding potential. Conductance (GNa) was calculated by the following equation: Gna = INa(Vm − Erev)(1)
where INa is the current amplitude at voltage Vm, and Erev is the Na^+^ reversal potential.

Steady-state activation curves were derived by plotting normalized GNa as a function of test potential and fitted using the Boltzmann equation:GNaGNa,Max = 1 + eVa-Emk(2)
where GNa,Max is the maximum conductance, Va is the membrane potential of half-maximal activation, Em is the membrane voltage, and k is the slope factor. For steady-state inactivation, normalized current amplitude (INa/INa,Max) at the test potential was plotted as a function of prepulse potential (Vm) and fitted using the Boltzmann equation:INaINa,Max = 1 + eVh-Emk(3)
where Vh is the potential of half-maximal inactivation, Em is the membrane voltage, and k is the slope factor. 

Transient INa inactivation decay was estimated using the standard exponential equation. The inactivation time constant (tau, *τ*) was fitted with the following equation:f(x) = A1e-t1 + C(4)
where A1 and ƒ1 are the amplitude and time constant, respectively. The variable C is a constant offset term along the Y axis. The goodness of fitting was determined by correlation coefficient (R), and the cutoff of R was set at 0.85.

To assess the effects on long-term inactivation (LTI), a four-step protocol was utilized, wherein cells underwent four 0 mV, 20 ms depolarization pulses separated by −90 mV, 40 ms recovery intervals. To standardize for differences in cell sizes, current densities were calculated by dividing the peak INa current amplitude by the membrane capacitance (Cm). The fraction of channels entering LTI was represented by normalizing the peak INa observed during depolarization cycles 2–4 to that observed during depolarization cycle 1 (INa/INa,Cycle 1), which was then plotted against the depolarization cycle. The cumulative (frequency-dependent) use dependency was assessed by administering 20 pulses with depolarization to −10 mV (50 ms duration) and 50 ms recovery intervals, with a train of 20 pulses at 10 Hz from a holding potential at −70 mV. The current pulses were normalized to the first recorded pulse, and the currents at the 2nd to 20th pulses were compared.

### 4.5. Statistics

One-way ANOVA followed by Tukey’s multiple comparison *t*-tests were used to analyze all electrophysiological data (*p*  <  0.05 was considered statistically significant; *n*  =  6–10 cells per group). The electrophysiological experiments employed a randomized-based design, and the analysis was not blinded. Normality was assessed, and the electrophysiological data sets displayed a normal distribution. No outliers were removed. EC_50_ and IC_50_ were calculated using the Sigmoidal dose–response (variable slope) fitting in GraphPad Prism.

## Figures and Tables

**Figure 1 ijms-25-08069-f001:**
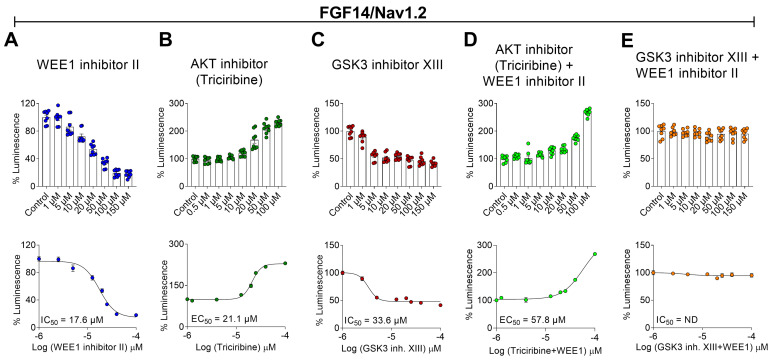
Evaluation of the effects of kinase inhibitors on the FGF14/Nav1.2 complex assembly using the in-cell split luciferase complementation assay (LCA). (**A**) Representative bar graph (top) and dose-response graph (bottom) of % luminescence reflecting the FGF14/Nav1.2 complex assembly in response to WEE1 inhibitor II (1–150 µM), (**B**) the AKT inhibitor (0.5–100 µM), and (**C**) GSK3 inhibitor XIII (1–150 µM). (**D**,**E**) are representative bar graphs (top) and dose responses (bottom) of % luminescence reflecting the FGF14/Nav1.2 complex assembly in response to WEE1 inhibitor II (15 µM) in combination with the indicated inhibitors. Percentage of luminescence (normalized to per plate control wells treated with 0.5% DMSO; *n* = 8 wells per plate) is plotted as a function of log concentration in the dose–response graphs at the bottom. Data are represented ± SEM. *Half maximal inhibitory concentration* (IC_50_) and half maximal effective concentration (EC_50_) were calculated using a sigmoidal dose–response fitting.

**Figure 2 ijms-25-08069-f002:**
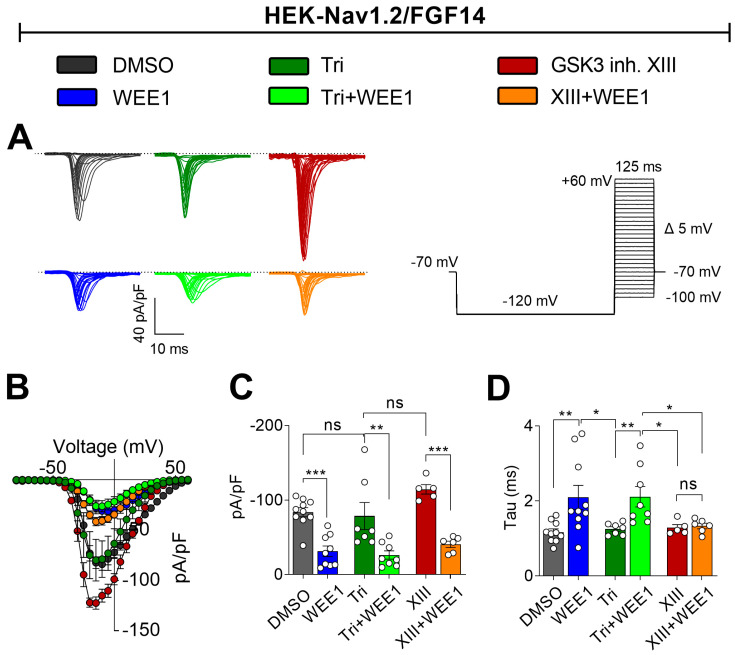
The interplay between WEE1 inhibitor II, triciribine, and GSK3 inhibitor XIII in regulating Nav1.2-mediated I_Na_. (**A**) Representative traces of I_Na_ from HEK293-Nav1.2 cells expressing FGF14-GFP (HEK-Nav1.2/FGF14). Traces were recorded in response to depolarizing voltage steps in the presence of WEE1 inhibitor II (WEE1; 10 µM), triciribine (tri; 10 µM), GSK3 inhibitor XIII (30 µM), and/or DMSO (0.01%); the voltage-clamp stimulating protocol is depicted on the right. (**B**) Current-voltage (I-V) relationships derived from the experimental groups are described in Panel (**A**). (**C**) Bar graphs representing peak current densities and (**D**) the time constant (tau) of fast inactivation of Nav1.2 channels from the experimental groups are described in Panel (**A**). Data are presented as mean ± SEM. Statistical significance is indicated as follows: * *p* < 0.05, ** *p* < 0.001, *** *p* < 0.0001, ns = nonsignificant, determined by one-way ANOVA followed by Tukey’s multiple comparisons test (*n* = 6–10). In this figure, GSK3 inhibitor XIII is referred to as XIII.

**Figure 3 ijms-25-08069-f003:**
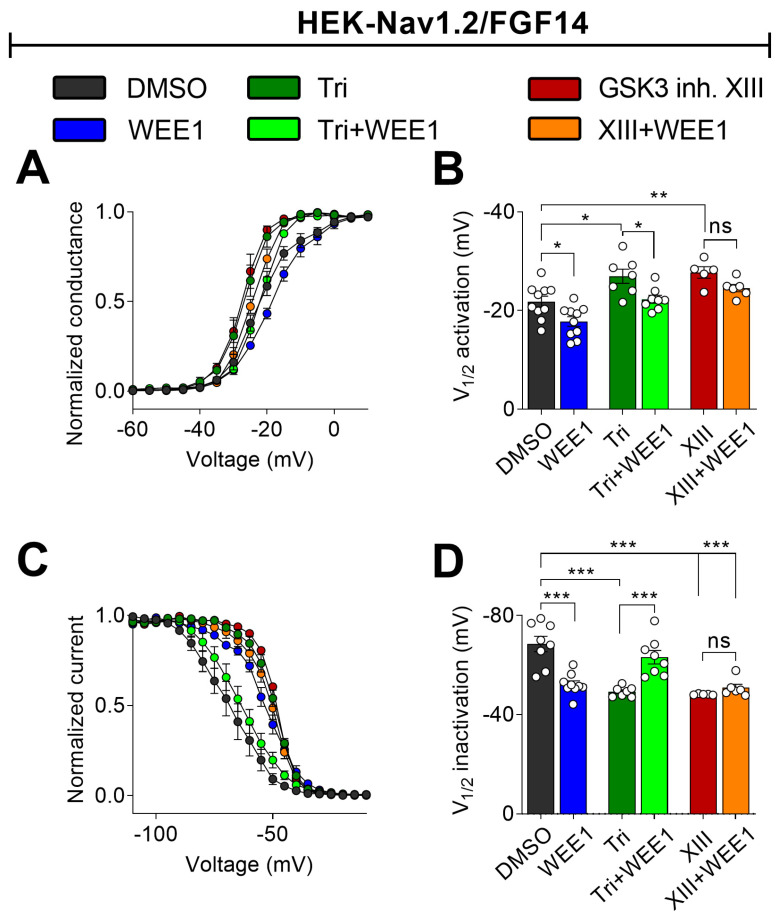
Synergy and competition between WEE1 inhibitor II, triciribine, and GSK3 inhibitor XIII in regulating Na_v_1.2 channel voltage dependence of activation and steady-state inactivation. (**A**) Normalized conductance plotted as a function of the voltage in HEK-Nav1.2/FGF14 cells treated with 0.1% DMSO or respective kinase inhibitors as shown with color-coded labels. The data were fitted with the Boltzmann function. (**B**) Bar graph summary of V_1/2_ of activation. (**C**) Voltage dependence of steady-state inactivation. (**D**) Bar graph summary of V_1/2_ of inactivation. Data are presented as mean ± SEM. Statistical significance is indicated as * *p* < 0.05, ** *p* < 0.001, *** *p* < 0.0001, ns = nonsignificant, determined by one-way ANOVA followed by Tukey’s multiple comparisons test (*n* = 6–10). In this figure, GSK3 inhibitor XIII is referred to as XIII.

**Figure 4 ijms-25-08069-f004:**
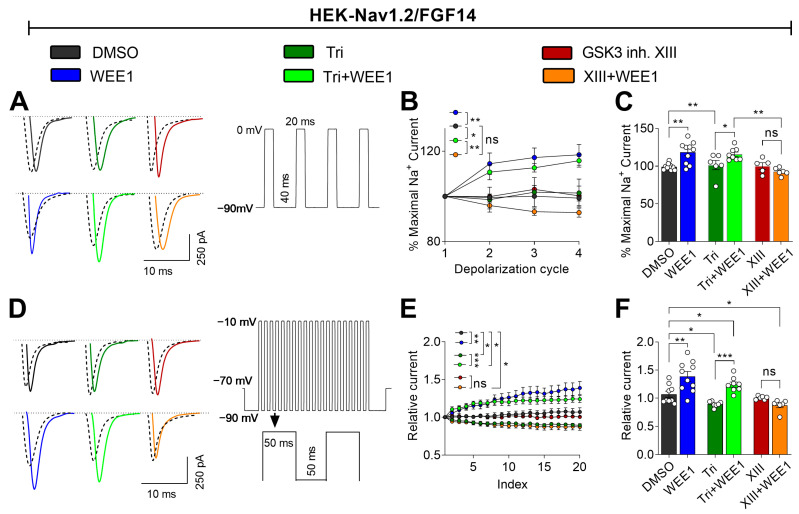
Crosstalk among WEE1 inhibitor, triciribine, and GSK3 inhibitor XIII on modulation of Na_v_1.2 channel long-term inactivation and cumulative inactivation properties. (**A**) Representative traces of *I*_Na_ elicited by HEK-Nav1.2/FGF14 cells from the indicated experimental groups in response to the depicted voltage-clamp protocol to induce long-term inactivation. (**B**) Long-term inactivation of Nav1.2 measured as channel availability versus depolarization. (**C**) Comparison of the relative *I*_Na_ amplitude at the 1st pulse to the 4th pulse for the indicated experimental groups. (**D**) Representative traces of *I*_Na_ elicited by HEK-Nav1.2/FGF14 cells from the indicated experimental groups in response to the depicted voltage-clamp protocol to induce use-dependent cumulative inactivation. (**E**) Characterization of cumulative inactivation of Nav1.2 channels induced by use dependency for the experimental groups described in (**D**). (**F**) Comparison of the relative *I*_Na_ amplitude at the 1st pulse to the 20th pulse for the indicated experimental groups. Data are presented as mean ± SEM. Statistical significance is indicated as * *p* < 0.05, ** *p* < 0.001, *** *p* < 0.0001, ns = nonsignificant, determined by one-way ANOVA followed by Tukey’s multiple comparisons test (*n* = 6–10). In this figure, GSK3 inhibitor XIII is referred to as XIII.

**Figure 5 ijms-25-08069-f005:**
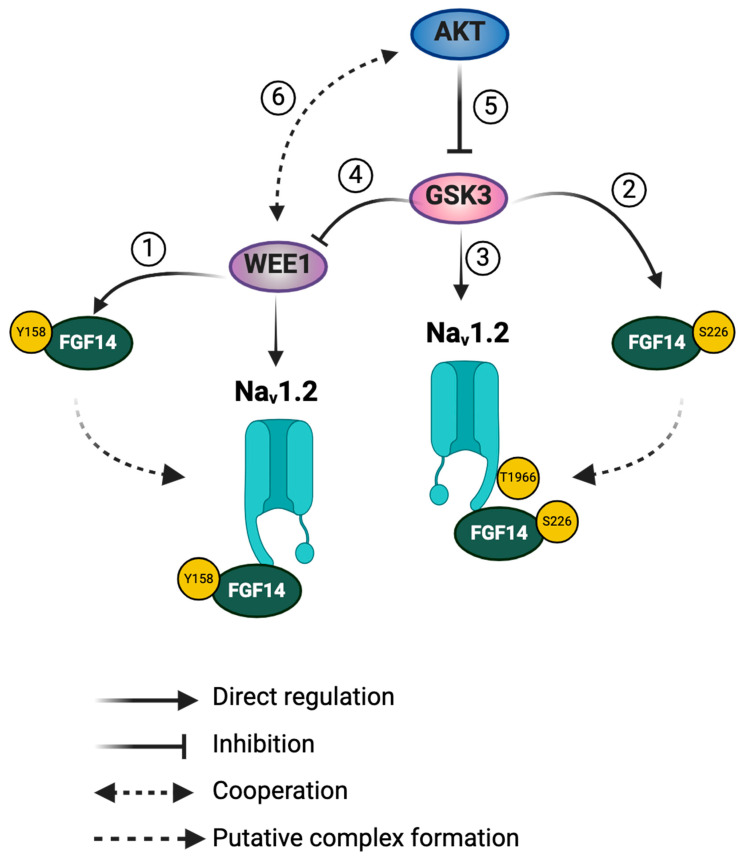
Putative crosstalk between WEE1 kinase, AKT, and GSK3 in regulating the Nav1.2/FGF14 signalosome. WEE1 kinase and GSK3β have been shown to directly regulate the FGF14/Nav1.2 complex assembly and its functional activity via phosphorylation of FGF14Y158 (1) and FGF14S226 (2), respectively. Additionally, GSK3β directly phosphorylates the Nav1.2 C-terminal tail at T1966 (3). Phosphorylation of FGF14Y158 by WEE1 kinase may increase its assembly with Nav1.2. Similarly, phosphorylation of FGF14S226 or Nav1.2T1966 by GSK3 may enhance the assembly of the FGF14/Nav1.2 complex. Moreover, GSK3 has been shown to degrade WEE1 kinase via ubiquitination (4), leading to a reduction in WEE1 kinase levels. There are no reports of direct phosphorylation of FGF14 or Nav1.2 by AKT. Therefore, AKT may influence the FGF14/Nav1.2 complex assembly and its functional activity indirectly through the suppression of GSK3β via inhibitory phosphorylation (5) or through a synergistic effect with WEE1 kinase (6).

**Table 1 ijms-25-08069-t001:** Nav1.2 currents in the presence of WEE1 inhibitor, triciribine, and/or GSK3 inhibitor XIII.

Condition	Peak Density	Activation	Steady-State Inactivation	Tau (τ)
	pA/pF	mV	mV	ms
DMSO	−83.85 ± 6.0 (10)	−21.0 ± 0.9 (10)	−68.5 ± 3.1 (8)	1.21 ± 0.07 (10)
WEE1 inh.	−31.34 ± 7.0 (9) ^#a^	−17.3 ± 1.0 (10) ^#e^	−55.14 ± 1.1 (9) ^#i^	2.1 ± 0.32 (10) ^#n^
Tri	−78.8 ± 18.16 (7) ^#ns^	−26.95 ± 1.4 (7) ^#f^	−49.2 ± 0.77 (7) ^#j^	1.25 ± 0.06 (7) ^#ns^
Tri + WEE1 inh.	−26.3 ± 5.5 (8) ^#%b^	−22.3 ± 0.8 (8) ^%g^	−63.1 ± 2.7 (11) ^%k^	2.12 ± 0.27 (8) ^#o^
GSK3 inh. XIII	−127.6 ± 7.2 (6) ^#c^	−27.75 ± 0.9 (6) ^#h^	−48.5 ± 0.4 (6) ^#l^	1.26 ± 0.07 (6) ^#ns^
GSK3 inh. XIII + WEE1 inh.	−40.7 ± 4.2 (6) ^#@d^	−24.5 ± 0.7 (6) ^@ns^	−50.8 ± 1.4 (6) ^#m@ns^	1.33 ± 0.07 (6) ^@ns^

Data are mean ± SEM; ns = nonsignificant; (*n*) = number of recordings; one-way ANOVA Tukey’s multiple comparisons test. # = DMSO % = Tri; @ = GSK3 inh. XIII. *^#^*^a^ *p* = 0.0008; ^#b^ *p* = 0.0003, ^%b^ *p* = 0.0041; ^#c^ *p* = 0.0292; ^#d^ *p* = 0.0330, ^@d^ *p* = 0.0001; ^#e^ *p* = 0.0184, ^#f^ *p* = 0.0015; ^%g^ *p* = 0.0163; ^#h^ *p* = 0.0001; ^#i^ *p* = 0.0002; ^#j^ *p* = 0.0001; ^%k^ *p* = 0.0008; ^#l^ *p* = 0.0001; ^#m^ *p* = 0.0001, ^#n^ *p* = 0.0103; ^#o^ *p* = 0.0013.

**Table 2 ijms-25-08069-t002:** Long-term inactivation and cumulative inactivation of Nav1.2 in the presence of WEE1 inhibitor, triciribine, and/or GSK3 inhibitor XIII.

Condition	LTI (% Maximal Na^+^ Current)		
	2nd Pulse	3rd Pulse	4th Pulse
DMSO	99.54 ± 1.6 (10)	100.1 ± 0.8 (10)	99.24 ± 1.9 (14)
WEE1	114.4 ± 4.7 (10) ^$a^	117.2 ± 4.4 (10) ^$d^	118.34 ± 4.6 (10) ^$g^
Tri	98.5 ± 5.5 (6) ^$ns^	101.86 ± 6.5 (6) ^$ns^	101.5 ± 6.1 (6) ^$ns^
Tri + WEE1	110.7 ± 3.4 (8) ^$b, %c^	112.65 ± 1.8 (8) ^$e, %f^	115.8 ± 2.8 (8) ^$h, %i^
GSK3 XIII	100.2 ± 1.0 (6) ^$ns^	103.7 ± 1.5 (6) ^$ns^	101.1 ± 3.6 (6) ^$ns^
GSK3 XIII + WEE1	95.94 ± 1.5 (6) ^$, @ns^	93.24 ± 1.6 (6) ^$, @ns^	92.82 ± 2.0 (6) ^$, @ns^
Cumulative inactivation (use dependency)			
Condition	10th Pulse	15th Pulse	20th Pulse
DMSO	1.03 ± 0.04 (9)	1.05 ± 0.04 (9)	1.07 ± 0.05 (9)
WEE1	1.26 ± 0.06 (10) ^$j^	1.31 ± 0.07 (10) ^$n^	1.4 ± 0.04 (10) ^$r^
Tri	0.92 ± 0.01 (7) ^$ns^	0.89 ± 0.02 (7) ^$ns^	0.89 ± 0.02 (7) ^$s^
Tri + WEE1	1.2 ± 0.05 (8) ^$k, %l^	1.22 ± 0.04 (8) ^$o, %p^	1.24 ± 0.01 (8) ^$t, %u^
GSK3 XIIIF	1.02 ± 0.01 (6) ^$ns^	1.0 ± 0.01 (6) ^$ns^	1.0 ± 0.04 (6) ^$ns^
GSK3 XIII + WEE1	0.89 ± 0.03 (6) ^$m, @ns^	0.87 ± 0.04 (6) ^$q, @ns^	0.87 ± 0.04 (6) ^$v, @ns^

Data are mean ± SEM; ns = nonsignificant; (*n*) = number of recordings; one-way ANOVA Tukey’s multiple comparisons test. $ = DMSO; % = Tri; @ = GSK3 inh. XIII. ^$a^ *p* = 0.0049; ^$b^ *p* = 0.0224; ^%c^ *p* = 0.0307; ^$d^ *p* = 0.0002; ^$e^ *p* = 0.0063; ^%f^ *p* = 0.0001; ^$g^ *p* = 0.0003; ^$h^ *p* = 0.0028; ^%i^
*p* = 0.0224; ^$j^ *p* = 0.0135; ^$k^ *p* = 0.0255; ^%l^ *p* = 0.0002; ^$m^ *p* = 0.0304; ^$n^ *p* = 0.0121; ^$o^ *p* = 0.0171; ^%p^ *p* = 0.0001; ^$q^ *p* = 0.0176; ^$r^ *p* = 0.0019; ^$s^ *p* = 0.0375; ^$t^ *p* = 0.0338; ^%u^ *p* = 0.0001; ^$v^ *p* = 0.0211.

## Data Availability

Data are contained within the article and Appendix A.

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
