# Peer review of "Crosstalk among WEE1 Kinase, AKT, and GSK3 in Nav1.2 Channelosome Regulation"

_ijms, 2024, doi:10.3390/ijms25158069_

Round 1

Reviewer 1 Report

Comments and Suggestions for Authors

In this manuscript, the authors evaluated the effect of inhibition of WEE1, GSK3, and AKT on Nav1.2 channelsome regulation. Using a luciferase-based assay, the authors discovered that individual or combined inhibition of WEE1, GSK3, and AKT by kinase inhibitors resulted in different effects on the assembly of FGF14/Nav1.2 complex. Following the same strategy, individual or combined application of WEE1 inhibitor II, triciribine, and XIII led to additive or counteracting effects on the electrophysiological properties of Nav1.2, which included Na+ current amplitude, fast inactivation, voltage dependence of activation, steady-state inactivation and long-term inactivation. The authors concluded that WEE1, GSK3, and AKT exerted both synergistic and hierarchical regulation of FGF14/Nav1.2 complex assembly and functions.

The experiments in this study were well executed and the manuscript was well written. However, this study did not provide a clear mechanism of WEE1 and GSK3/AKT-mediated regulation of FGF14/Nav1.2. The following issues need to be addressed:

1. In the putative model and discussion, the authors stated that WEE1 kinase regulated FGF12/Nav1.2 complex assembly and function via phosphorylation of FGF14Y158. In a previous study (PMID 34831326), the FGF14Y158A mutation alone, which abolished the putative phosphorylation site of WEE1, did not significantly decrease FGF14/Nav1.2 assembly or peak density. This contradiction requires further explanation.

2. Individual application of either WEE1 or GSK3 inhibitor both reduced FGF14/Nav1.2 complex assembly. However, the combined application of WEE1 and GSK3 inhibitors canceled each other’s effect. An explanation or discussion of these results is needed.

3. To dissect the regulatory pathway of WEE1 and GSK3/AKT on FGF14/Nav1.2 channelsome, data of triciribine and XIII treatment on FGF14Y158A/Nav1.2 should be provided.

Author Response

Rev 1:

In this manuscript, the authors evaluated the effect of inhibition of WEE1, GSK3, and AKT on Nav1.2 channelsome regulation. Using a luciferase-based assay, the authors discovered that individual or combined inhibition of WEE1, GSK3, and AKT by kinase inhibitors resulted in different effects on the assembly of FGF14/Nav1.2 complex. Following the same strategy, individual or combined application of WEE1 inhibitor II, triciribine, and XIII led to additive or counteracting effects on the electrophysiological properties of Nav1.2, which included Na+ current amplitude, fast inactivation, voltage dependence of activation, steady-state inactivation and long-term inactivation. The authors concluded that WEE1, GSK3, and AKT exerted both synergistic and hierarchical regulation of FGF14/Nav1.2 complex assembly and functions.

The experiments in this study were well executed and the manuscript was well written. However, this study did not provide a clear mechanism of WEE1 and GSK3/AKT-mediated regulation of FGF14/Nav1.2. The following issues need to be addressed:

  1. In the putative model and discussion, the authors stated that WEE1 kinase regulated FGF12/Nav2 complex assembly and function via phosphorylation of FGF14Y158. In a previous study (PMID 34831326), the FGF14Y158Amutation alone, which abolished the putative phosphorylation site of WEE1, did not significantly decrease FGF14/Nav1.2 assembly or peak density. This contradiction requires further explanation.

Response from authors: We apologize for any lack in clarity. However, as indicated in the original manuscript we have previously published that FGF14 suppresses Nav1.2-mediated peak current density in HEK-Nav1.2 cells, a phenotype that is absent in HEK-Nav1.2 cells expressing FGF14Y158A (Table 1, PMIDW 34831326).

  1. Individual application of either WEE1 or GSK3 inhibitor both reduced FGF14/Nav2 complex assembly. However, the combined application of WEE1 and GSK3 inhibitors canceled each other’s effect. An explanation or discussion of these results is needed.

Response from authors: We have clarified this in the discussion in line 316-317.

  1. To dissect the regulatory pathway of WEE1 and GSK3/AKT on FGF14/Nav2 channelsome, data of triciribine and XIII treatment on FGF14Y158A/Nav1.2 should be provided.

Response from authors: We have conducted a new set of experiments using the in-cell split-luciferase assay (LCA) to reconstitute FGF14Y158A/Nav1.2 assembly in the presence of WEE1 and GSK3/AKT inhibitors. The new figure for this data is included in the revised manuscript as Supplementary Fig. S1.

Reviewer 2 Report

Comments and Suggestions for Authors

The manuscript is complicated and difficult to read.

In line 11 it is not specified what AKT/GSK3 mean.

In line 73 it is written: aiding in the development of targeted therapies for these conditions.    How will the results presented in this study aid in the development of targeted therapies for neurodevelopmental disorders ?

Transient sodium currents should fully inactivate in time. In Figure 2 brown traces and blue traces do not fully inactivate. Why is it so ?

In Figure 2A (lower panel) the currents are weakly visible, please change the color.

Current amplitudes (Fig 2) are shown in pA/pF and not in pA , why is it so ?

In Fig 2B the holding potential is -120 mV whereas in line 408 it is written that the holding potential is -70 mV

In Fig 3D inactivation curves are shown, please show the recordings of steady-state inactivation.

In Fig 2B activation protocol is shown, please show steady-state inactivation protocol in Fig 3D.

In Fig 2B activation protocol is shown, please show the protocols of LTI and cumulative inactivation in Fig 4.

Fig 4 B and 4 E show long term inactivation and cumulative use-dependency, respectively. Both protocols should progressively decrease current amplitudes. In Fig 4B and E, however, blue curves show increase in current amplitudes. Why is it so ?

In line 135 and in all other lines where p is indicated please indicate which statistical test was used.

In line 135 it is written DMSO: −83.85 ± 6.0 pA/pF whereas in lines 140 and 143 it is written FGF14-GFP: −83.85 ± 6.0 pA/pF why is it so ?

In line 141 p value is indicated twice

This study shows that three kinase inhibitors influence current density and kinetic properties of Nav1.2 channels. Please explain in the discussion section the physiological significance of these findings. For example, activation curves are shifted as compared to control (Fig 3C) , what is the physiological significance of this phenomenon ?

In lines 357 and 358 SCN2A channelopathies are mentioned. What are these channelopathies ?

Author Response

Rev 2:

The manuscript is complicated and difficult to read.

Response from authors: We have added new data, streamlined some of the text and included new references.

In line 11 it is not specified what AKT/GSK3 mean.

Response from authors: We have clarified that AKT/GSK3 are kinases.

In line 73 it is written: aiding in the development of targeted therapies for these conditions.    How will the results presented in this study aid in the development of targeted therapies for neurodevelopmental disorders?

Response from authors: We have clarified this point in line 78.

Transient sodium currents should fully inactivate in time. In Figure 2 brown traces and blue traces do not fully inactivate. Why is it so ?

Response from authors: The effect observed on transient Na+ current inactivation is a phenotype produced by the respective treatment. In Figure 2A (lower panel) the currents are weakly visible, please change the color.

Response from authors: We changed the colors as suggested.

Current amplitudes (Fig 2) are shown in pA/pF and not in pA; why is it so ?

Response from authors: To normalize responses on cell size, we report current densities in pA/pF, and that is what we depict in our illustrations.

In Fig 2B the holding potential is -120 mV whereas in line 408 it is written that the holding potential is -70 mV

Response from authors: Yes, the holding potential is -70 mV, please check the protocol shown in Fig. 2A (inset).

In Fig 3D inactivation curves are shown, please show the recordings of steady-state inactivation.

Response from authors: We have added the recording traces along with protocol for the steady-state inactivation in Supplementary Fig S2.

In Fig 2B activation protocol is shown, please show steady-state inactivation protocol in Fig 3D.

Response from authors: We have added the recording protocol for the steady-state inactivation in Supplementary Fig S2.

In Fig 2B activation protocol is shown, please show the protocols of LTI and cumulative inactivation in Fig 4.

Response from authors: We have added the recording protocol for the LTI and cumulative inactivation in Fig 4 respective panels (inset).

Fig 4 B and 4 E show long term inactivation and cumulative use-dependency, respectively. Both protocols should progressively decrease current amplitudes. In Fig 4B and E, however, blue curves show increase in current amplitudes. Why is it so ?

Response from authors: Both long-term inactivation (LTI) and cumulative use-dependency showed FGF14 mediated effect on Nav1.2 channel as previously reported (PMID 34831326). The blue traces (now dark blue) showed the effects of WEE1 inhibitor, indicating that the WEE1 inhibitor rescues the effect of FGF14 for LTI and cumulative use-dependency.

Round 2

Reviewer 1 Report

Comments and Suggestions for Authors

No further suggestion.